# SARS-CoV-2 Spike Protein-Expressing Enterococcus for Oral Vaccination: Immunogenicity and Protection

**DOI:** 10.3390/vaccines11111714

**Published:** 2023-11-14

**Authors:** Alexander Suvorov, Svetlana Loginova, Galina Leontieva, Tatiana Gupalova, Yulia Desheva, Dmitry Korzhevskii, Tatiana Kramskaya, Elena Bormotova, Irina Koroleva, Olga Kopteva, Olga Kirik, Veronika Shchukina, Sergey Savenko, Dmitry Kutaev, Sergey Borisevitch

**Affiliations:** 1Scientific and Educational Center, Molecular Bases of Interaction of Microorganisms and Human of the World-Class Research Center, Center for Personalized Medicine, FSBSI, IEM, 197022 Saint-Petersburg, Russia; alexander_suvorov1@hotmail.com (A.S.); galeonte@yandex.ru (G.L.); tvgupalova@rambler.ru (T.G.); dek2@yandex.ru (D.K.); tatyana.kramskaya@gmail.com (T.K.); bormotovae@rambler.ru (E.B.); ivkoroleva@yandex.ru (I.K.); olga.s.kopteva@yandex.ru (O.K.); olga_kirik@mail.ru (O.K.); 2Federal State Budgetary Institution 48th Central Research Institute of the Ministry of Defense of the Russian Federation, 141306 Moscow, Russia

**Keywords:** recombinant probiotic-based vaccines, probiotic strain *E. faecium* L3, mucosal vaccines, SARS-CoV-2 spike protein, vaccine efficacy

## Abstract

The declaration of the conclusion of the COVID-19 pandemic notwithstanding, coronavirus remains prevalent in circulation, and the potential emergence of novel variants of concern introduces the possibility of new outbreaks. Moreover, it is not clear how quickly and to what extent the effectiveness of vaccination will decline as the virus continues to mutate. One possible solution to combat the rapidly mutating coronavirus is the creation of safe vaccine platforms that can be rapidly adapted to deliver new, specific antigens in response to viral mutations. Recombinant probiotic microorganisms that can produce viral antigens by inserting specific viral DNA fragments into their genome show promise as a platform and vector for mucosal vaccine antigen delivery. The authors of this study have developed a convenient and universal technique for inserting the DNA sequences of pathogenic bacteria and viruses into the gene that encodes the pili protein of the probiotic strain *E. faecium* L3. The paper presents data on the immunogenic properties of two *E. faecium* L3 vaccine strains, which produce two different fragments of the coronavirus S1 protein, and provides an assessment of the protective efficacy of these oral vaccines against coronavirus infection in Syrian hamsters.

## 1. Introduction

Over the past two decades, there has been ongoing surveillance and study of various coronavirus variants within the human population [1,2,3,4,5]. Traditionally, weakly virulent human coronaviruses have been responsible for causing a range of seasonal acute respiratory viral infections. However, in recent years, coronaviruses have been implicated in the emergence of sporadic acute respiratory syndromes characterized by alarmingly elevated mortality rates, exemplified by SARS-CoV and MERS-CoV. The emergence of SARS-CoV-2 has initiated a global pandemic, subsequently expedited the development of coronavirus vaccines, and emphasized the necessity for the prompt formulation and implementation of measures to mitigate severe viral and bacterial infections.

Over the past three years since the outbreak of the SARS-CoV-2 pandemic, various vaccines based on traditional and innovative technologies have been developed (https://www.who.int/publications/m/item/draft-landscape-of-COVID-19-candidate-vaccines, accessed on on 1 May 2022). These include subunit protein vaccines, RNA vaccines, replicating and non-replicating viral vector vaccines, inactivated vaccines, DNA vaccines, vaccines on the base of virus-like particles, and live attenuated vaccines. As of 2022, a total of 11 vaccines, including inactivated vaccines, viral vector vaccines, mRNA vaccines, and subunit protein vaccines, have been approved by the World Health Organization for emergency use (https://extranet.who.int/pqweb/vaccines/vaccinesCOVID-19-vaccine-eul-sissed, accessed on 20 May 2022).

Three years following the onset of the pandemic, a retrospective analysis of excess mortality reveals that countries implementing well-established vaccine preventive measures exhibited the lowest excess mortality rates [6].

The safety of all vaccines approved by the WHO has been proven in clinical trials. The severity of post-COVID complications far exceeds the incidence of vaccination-related side effects. Thus, global retrospective studies conducted using clinical data from millions of both vaccinated and unvaccinated individuals have demonstrated that, on average, the relative risk of developing cardiovascular complications for infected individuals compared to the vaccinated group was sevenfold higher [7].

Although, on average, post-vaccination complications are relatively infrequent, they do indeed occur. The following adverse effects have been documented after COVID-19 vaccine administration: thrombotic thrombocytopenia [8,9], myocarditis [10,11], allergic reactions [12], autoimmune hepatitis, thyroid dysfunction [13,14], neurological disorders [15,16], and other [17,18,19].

In addition to vaccine side effects in the general population, there is also a risk of adverse reactions in high-risk groups, such as the elderly, people with immune diseases and AIDS, transplant recipients, and cancer patients [20].

Despite the undeniable effectiveness of existing vaccines in preventing coronavirus, it is important to work towards improving existing vaccines platforms and developing safer vaccine candidates.

Using recombinant probiotic bacterial strains to deliver vaccine antigens through the gastrointestinal mucosa could be a safe and effective vaccine strategy.

The resistance of animals to infection after oral administration of a vaccine agent was described over a hundred years ago [21,22]. This research led to the theory of local immunity [23,24], which develops independently from the systemic immune response. Contemporary concepts of mucosal immunity do not deny its connection with the systemic immune response and suggest that local immune reactions play a crucial role in protection against infection at the main entry routes of viruses, including the respiratory, alimentary, and urogenital tracts, the outer surface of the eye, and the skin. However, the respiratory and alimentary routes are the most significant [25,26].

Several successful mucosal vaccines exist today, such as vaccines against cholera, polio, influenza, and other infections [27,28,29]. Mucosal immunity is compartmentalized [30,31,32], but a connection between different mucosal sites does exist. An immune response in one mucosal area can result in more or less pronounced immune reactions in other mucosal sites [33,34,35]. Recent evidence suggests that both local and systemic specific immune responses can be stimulated when vaccine antigens reach the mucous membranes as part of probiotic bacteria through intranasal, vaginal, or oral administration [36,37,38,39].

In our current study, we evaluated two variations of a recombinant probiotic vaccine candidate delivered orally, based on the L3 *E. faecium* strain. The development principles of these probiotic-based vaccine candidates have been previously outlined [40]. We evaluated their immunogenic properties and protective efficacy against coronavirus infection in hamsters.

## 2. Materials and Methods

### 2.1. Cell Culture

Vero C1008 cells (obtained from ECACC: 85020206) were used in cell culture experiments. These cells were cultured in Minimum Essential Medium (MEM) (supplied by PanAco, Moscow, Russia) that was supplemented with 2% fetal calf serum (from Sigma, Saint Louis, MO, USA) in 225 cm^2^ cell culture flasks (manufactured by Cellstar, Greiner Bio-One GmbH, Frickenhausen, Germany).

The Vero cell monolayers were passaged every 3–4 days by trypsinization, with the split ratio being approximately 1:4.

### 2.2. The Virus

The SARS-CoV-2 virus strain used in the study was hCoV-19/Russia/SAB-1502/2021 (belonging to the South Africa/gamma 1.351 501 V2 lineage with the substitutions S D80A, D215G, E484K, N501Y), sourced from the Federal Budgetary Research Institution-State Research Center of Virology and Biotechnology “VECTOR”, Rospotrebnadzor. The selection of the virus was grounded in the observation that, during that specific period, both the previous Beta variant and the emerging Gamma variant of coronavirus were concurrently in circulation.

#### 2.2.1. Assessment of Infectious Activity

Titration of the SARS-CoV-2 virus was conducted using both a plaque assay in Vero cell culture monolayers under solid overlay media [41] and a TCID50 assay in Vero cell culture monolayers [42].

#### 2.2.2. Preparation of Virus for Challenge

Prior to the challenge, the SARS-CoV-2 virus was propagated in Vero cell culture. The cell monolayer was produced by cultivating 2 × 10^5^/mL cells in plastic flasks (Cellstar, Greiner Bio-One GmbH, Germany) for 24 h with 5% CO_2_ in a growth medium consisting of a modified Eagle’s medium under observation using a light microscope (MEM, Gibco, Thermo Scientific, Waltham, MA, USA ) supplemented with 2% FBS (Gemini, Sacrament, CA, USA), 100 μg/mL penicillin, and 100 μg/mL streptomycin. The virus was adsorbed onto the cell monolayer at a concentration of 1PFU/mL for 60 min at 37.0 °C. After the adsorption, the inoculate was removed, the cells were washed with MEM medium, and 7–8 mL of fresh growth medium was added. The flasks were incubated at 5% CO_2_ and 37.0 °C for 48 h. The cells were pelleted, the precipitate was separated by centrifugation, and the supernatant was aliquoted and stored at −70 °C. The virus-containing product was evaluated for its sterility and infectious activity. The infectious activity of SARS-CoV-2 was assessed using plaque and TCID_50_ assays.

The following outlines the characteristics of the SARS-CoV-2 virus as it pertains to infection in golden Syrian hamsters (Table 1).

### 2.3. Bacteria

The L3 strain of Enterococcus faecium and the DH5α, M15, and BL21 strains of Escherichia coli were obtained from the Institute of Experimental Medicine’s collection. The *E. coli* and *E. faecium* strains were grown in Luria Bertani (LB) medium (Oxoid, Nepean, ON, Canada) or Todd Hewitt Broth (THB) (HiMedia, Maharashtra, India) at 37 °C with constant shaking for 14 h. LB agar (Lennox L agar, Thermo Fisher Scientific, Waltham, MA, USA) and Enterococcus Differential Agar Base (TITG Agar Base) (HiMedia, Maharashtra, India) without antibiotics or with 10 μg/mL erythromycin were used as solid media for the cultivation, quantification, and identification of the bacteria and erythromycin-resistant enterococcal transformants.

### 2.4. Animals

Syrian golden hamsters weighing between 50 to 60 g were obtained from the ‘Andreevka’ Branch of the Federal State Budgetary Scientific Institution “Scientific Center for Biomedical Technologies of the Federal Medical and Biological Agency” (FSBSN NTBMT FMBA Branch ‘Andreevka’).

The animals were housed in a barrier facility under appropriate conditions in accordance with standard procedures, with a 3-day acclimation period.

The animals were maintained and handled according to the guidelines of SP №1045 73 for vivarium arrangement (GOST R 53434-2009), the “Guidelines for laboratory animals.” (M. 2010), and animal protection regulations.

The hamsters were fed a complete pelleted diet and housed in plastic cages with sawdust from wood and REHOFIX^®^ bedding (Rosenberg, Germany) used as bedding material. The temperature and humidity were maintained between 15 and 21 °C and 30–70%, respectively, with a 12 h light/12 h darkness cycle.

### 2.5. Animal Procedures

Before immunization, the Syrian golden hamsters were evaluated for behavior, appetite, hair quality, and mucous membrane health. The animals were weighed for grouping purposes and divided into five experimental groups.

The experimental design is depicted in Figure 1.

The experimental samples of live recombinant probiotic vaccine candidates Vac1 and Vac2, as well as the recipient strain *E. faecium* L3, were orally administered in a 0.1 mL PBS suspension at a dose of 1 × 10^9^ CFU. During the first round of vaccination, the vaccine was administered once daily for three consecutive days. The second round was administered in a similar manner three weeks later.

Twenty-eight days after the second immunization, blood and swab samples (from the throat and cheek) were collected from five animals per group to assess the immunogenicity of the vaccine candidates. The blood was taken under anesthesia from the subclavian vein, and the animals were euthanized by cervical dislocation.

Two days later, hamsters were given an oral dose of 200 μL of SARS-CoV-2 virus at a dose of 4.3 log PFU. On days 3 and 6 post-inoculation, five hamsters per group were euthanized by cervical dislocation, and lung samples were collected to assess virus replication. The tissues were homogenized in PBS containing 100 μg/mL penicillin and 100 μg/mL streptomycin. The resulting 10% homogenates were analyzed by plaque assay on Vero cell cultures under solid overlay media. Lung tissue samples were also obtained from three additional hamsters per group for each day for histological examination.

The clinical examinations were performed during the infection process to monitor for deviations from normal physiology. The experiments were conducted in accordance with EU Directive 2010/63/EU (https://eur-lex.europa.eu/LexUriServ/LexUriServ.do?uri=OJ:L:2010:276:0033:0079:en:PDF, accessed on 2 November 2021…) for the use of animals in scientific research and the Federation of European Laboratory Animal Science Associations (FELASA) Recommendations for the health monitoring of mouse, rat, hamster, guinea pig, and rabbit colonies in breeding and experimental units. The experiments were approved and carried out in accordance with these guidelines and under the supervision of the local biomedical ethics committee (as recorded in meeting minutes dated 3 November 2021).

### 2.6. ELISA Test

The commercial trimeric full-length WT S protein of SARS-CoV-2 (Vector-Best, Novosibirsk, Russia) and recombinant SA and SB proteins, which are analogues of coronavirus inserts in the Vac1 and Vac2 vaccine strains, respectively, were used as antigens in the ELISA assay. The ELISA was performed as described by Gupalova et al. [2]. Maxisorb 96-well plates (Nunc; Sjælland, Denmark) were coated overnight at 4 °C with 0.25 μg/mL of SA and SB proteins in a 0.1 M sodium carbonate buffer with pH 9.3. A series of two-fold dilutions of the sample (100 μL) was added to duplicate wells and incubated for 1 h at 37 °C. The plates were washed with a blocking buffer (0.05% Tween-20 in PBS) between each stage. The same buffer was used for dilution of the serum and reagents. HRP-labeled goat anti-hamster IgA or IgG antibodies (Sigma) were added (100 μL/well). After incubation at 37 °C for 1 h, the plates were developed with 100 μL/well TMB substrate (BD Bioscience). A color reaction was detected after 20 min of incubation, which was stopped with 30 μL of 50% sulfuric acid. The endpoint ELISA titers were expressed as the highest dilution that yielded an optical density at 450 nm (OD450) greater than the mean OD450 plus 3 standard deviations of the negative control wells.

### 2.7. Evaluation of Virus Neutralizing Activity of Serum and Swabs

#### 2.7.1. Plaque Reduction Neutralization Test on Vero Cell Culture Monolayers under Solid Overlay Media

The presence of antibodies that neutralize the replication of 100 PFU/mL of SARS-CoV-2 in Vero 1008C cells was determined through a plaque assay in Vero cell culture monolayers under solid overlay media. Two-fold dilutions of heat-inactivated hamster sera were tested, with four replicates per each dilution. The viral cytopathic effect was evaluated on day 4, and the highest dilution of blood serum in which the decrease in plaque number exceeded the negative control by at least 50% was defined as the antibody titer.

#### 2.7.2. The Neutralizing Properties of a Sample Were Evaluated through the Inhibition of Binding between SARS-CoV-2 Protein S1 and Human ACE2 in an ELISA Assay

Micro well plates were coated with 100 μL of ACE2 (HyTest, Moscow, Russia) at a concentration of 1.6 μg/mL in 0.01 M phosphate-buffered saline (PBS) solution at pH 7.4, and incubated for 24 h at 4 °C. In parallel, 150 μL of horseradish peroxidase-conjugated recombinant full-length S glycoprotein of the Wuhan-Hu-1 SARS-CoV-2 virus (ID: 43740568) was added to the wells of another microtiter 96-well plate at a pre-selected optimal concentration. Whole animal serum samples and nasopharyngeal washes were added to the wells in a 1:16 dilution, followed by incubation for 15 min at 37 °C with shaking.

The ACE2-coated plate was washed three times with 350 μL of PBS with 0.05% Tween-20 (Sigma-Aldrich, Germany). The co-cultured test samples with the SARS-CoV-2 S-protein conjugate was transferred to the ACE2-coated wells, incubated for 1 h at 37 °C with shaking, and then washed five times with 350 μL of PBS with 0.05% Tween-20. After an additional incubation for 1 h at 37 °C, the plates were developed with 100 μL/well TMB substrate (BD Bioscience) and a color (OD450) was detected after 25 min of incubation, followed by stopping the reaction with 30 μL of 50% sulfuric acid.

A Syrian golden hamster serum that was free of specific antibodies to SARS-CoV-2 was used as a negative control, while human serum with a predetermined concentration of IgG to the SARS-CoV-2 S protein was used as a positive control.

The Index of Neutralization (IN) was determined by the formula:IN = 100 − (ODs/ODnc) × 100 (%),
where ODs represents the mean OD450 value in the wells containing the test sample, and ODnc represents the mean OD450 value in the wells containing the negative control.

A neutralization index of greater than 20% was considered a positive result and was consistent with the IN observed in the positive control, which had a known neutralizing antibody concentration of 12.5 PFU/mL.

### 2.8. Evaluation of the Antiviral Efficacy of Experimental Samples

Evaluation was Conducted According to the Guidelines Set by the Scientific Centre for Expert Evaluation of Medicinal Products of the Ministry of Health of the Russian Federation

The viral inhibition coefficient (CI, %) was calculated using the following formula:CI = [(Anc − As)/Anc] × 100(%)
where Anc is the concentration of the virus determined by the plaque assay in Vero cell culture monolayers under solid overlay media in the absence of test samples (PFU/mL) and As is the concentration of the virus determined by the plaque assay in Vero cell culture monolayers under solid overlay media after the addition of test samples (PFU/mL).

### 2.9. Bioinformatics Analyses

DNA and putative protein analysis were performed employing BLAST NCBI (http://blast.ncbi.nlm.nih.gov/Blast.cgi, accessed on 1 December 2021) and ExPASy (http://www.expasy.org program, accessed on 20 December 2021) packages available in public domains. DNA primer design was accomplished by the Primer 3.0 computer program. Protein sequence analysis for the presence of the B-cell and T-cell epitopes was performed by employing the free Immune Epitope Database and Analysis Resource (IEDB) (Appendix A).

### 2.10. Construction of Recombinant Probiotic Vaccines Vac1 and Vac2

Two vaccine strains were generated by incorporating DNA fragments encoding two distinct S1 protein regions of coronavirus into the genome of the probiotic strain *E. faecium* L3 (Figure 2).

Schematic representation of the domain arrangement of the SARS-CoV2 S1 protein. N-terminal domain, RBD-receptor-binding domain, RBM-receptor-binding motif, and SD1 and SD2-subdomains.

The diagram above shows the relative position of the SA and SB coronavirus amino acid sequences inserted in the vaccine strains Vac1 and Vac2, respectively.

The construction of clone Vac1 was previously reported [40]. The Vac2 clone was produced using the same method, with the only difference being the incorporation of a different DNA sequence (GenBank: ON803610.1) into the integrative plasmid that was inserted into the chromosome of *E. faecium* L3 (Table 2). This sequence encodes the receptor-binding domain of SARS-CoV-2, including three amino acid mutations relative to the wild-type virus: K417N, E484K, and N501Y.

### 2.11. Recombinant Proteins Production

The production of recombinant proteins SA and SB, derived from SARS-CoV-2 RNA fragments cloned into *E. coli* expression vectors, was carried out using a previously reported method [40]. The gene fragments of the S1 spike protein were inserted into *E. coli* using the primers and vectors listed in Table 3.

The *E. coli* strains producing SA and SB were grown in Terrific broth supplemented with 25 µg/mL kanamycin and 100 µg/mL ampicillin, respectively, until reaching the late logarithmic growth phase (OD 600 = 0.7 to 0.9). Protein expression was induced by the addition of IPTG, and the cells were cultured for an additional 4.5 h. The cells were then collected by centrifugation and stored at −70 °C. After thawing, the cell pellet was suspended in 8 M urea, 0.1 M Na_2_HPO_4_, and 0.1 M NaH2PO4 (pH 8.0), and incubated at room temperature while stirring for 1 h. The resulting supernatant was purified through a Ni Sepharose column (Qiagen, Hilden, Germany), the proteins were eluted using 0.4 M imidazole under denaturing conditions, dialyzed against 0.4 M NaCl, 0.02 M Na_2_HPO_4_/NaOH (pH 9.2) overnight at 6 °C without stirring, filtered through 0.45-micron Millipore filters, and stored at 6 °C. The molecular weight of protein SA was found to be 24.5 ± 0.5 kDa, while protein SB had a molecular weight of 22.5 ± 0.5 kDa (Appendix A). The amino acid sequence of the polypeptides was confirmed using MALDI-TOF/TOF analysis (Bruker Daltonics, Bremen, Germany) and was found to correspond to the structure of the nucleotide inserts in the *E. coli* genome.

### 2.12. Macroscopic Examination of the Lung

During the visual inspection, the following features were documented: color, surface characteristics, consistency, airway assessment, presence of exudate or effusion. The pathological alterations in the lung tissue due to the SARS-CoV-2 virus were categorized into 5 classes based on the extent of harm and quantified using a score ranging from 0 to 4 as per the defined criteria.

The scoring system is outlined in Appendix A.

### 2.13. Histological Analysis

After harvesting, the animal lungs were preserved in 10% neutral formalin for 21 days at ambient temperature to inactivate the virus. After fixation, samples were rinsed three times in distilled water, one hour for each change. The material underwent dehydration using ethanol solutions of increasing concentration and embedded in paraffin (Richard-Allan Scientific Paraffin, Microm, Walldorf, Germany) with Spin Tissue Processor STP 120 (Microm, Germany) following established protocol. Then, 5 μm thick sections of the right lung paraffin block were cut using a rotary microtome (Rotary 3003 PFM Medical, PFM Medical, Cologne, Germany) and mounted on HistoBond m adhesive slides (Marienfeld, Lauda-Königshofen, Germany). The lung tissue was assessed via hematoxylin-eosin staining, resulting in blue-violet staining of the cell nuclei, moderate oxyphilic cytoplasm in the smooth muscle cells of blood vessel walls, visible red-brown erythrocytes in the vessel lumen, and pink connective tissue fibers. The specimens were examined under a microscope (Leica DM750) and photographs were captured using a digital camera (Leica ICC50, Leica, Wetzlar, Germany) with standardized light, contrast, and magnification settings.

### 2.14. Statistical Analyses

Data normality was tested by Shapiro–Wilk test, and a student’s t test was performed to obtain the statistical significance (*p*-value). The results are presented as the mean ± SEM. Statistically significant differences between groups were determined by ANOVA with Tukey’s multiple comparison test or in the case of non-normally distributed data, a nonparametric Mann–Whitney U-test. Data were analyzed with the statistical module of GraphPad Prism 6 software (GraphPad Software, Inc., San Diego, CA, USA). *p* values of <0.05 were considered significant.

Pearson’s correlation was used to determine the correlation between antigen-specific IgG and IgA in matched saliva and serum/plasma samples collected from the same person at the same time point.

## 3. Results

The objective of the study was to assess the protective efficacy of two recombinant enterococcal strains designated as Vac1 and Vac2 as probiotic vaccines.

The construction of Vac1 was previously described [40], and Vac2 was constructed using the same approach.

A 533 bp S1 DNA fragment encoding the complete receptor-binding domain (RBD) region of the SARS-CoV-2 virus was selected from the database (GenBank: ON803610.1) for insertion into the probiotic genome. The target protein was analyzed using ExPASy and IEDB tools, revealing the presence of linear B cell determinants and MHCI and MHCII T cell epitopes.

Purified recombinant proteins homologous to the insert were generated to analyze the specific immune response to the viral polypeptides expressed by the probiotic bacterium. A recombinant protein designated as SB was generated for the Vac2 strain, with a molecular weight of 22.5 kDa (Appendix A). The homology of the amino acid sequence of the protein SB to the target DNA fragment was confirmed using MALDI-TOF/TOF analysis.

A recombinant protein designated as SA was previously generated for the Vac1 strain and described [40].

### 3.1. The Safety and Immunogenic Properties of the Vaccine Strains Vac1 and Vac2

Three experimental groups were established: two received the recombinant vaccine strains Vac1 and Vac2 and one served as a control group and received the unmodified *E. faecium* L3 strain.

Physical assessments were performed on Syrian golden hamsters during the vaccination process, including weight measurements and evaluations of behavior, body fatness, hair, skin, mucous membranes, excrement, and breathing. All observations were found to be within normal physiological ranges.

The body weight of hamsters in all experimental groups did not show significant differences, which indicated normal physiological conditions and the absence of toxic effects from the administered probiotic dose (Table 4).

Blood serum and nasopharyngeal swab samples were collected 28 days after the second round of immunization.

The humoral immune response was evaluated using three methods. The neutralizing antibody levels were measured through plaque reduction neutralization testing on Vero cell monolayers in solid overlay media and an enzyme-linked immunosorbent assay (ELISA) utilizing human angiotensin-converting enzyme 2 (ACE-2). The presence of specific IgG antibodies was also determined through ELISA, using full-length S1 protein from a commercial kit, as well as recombinant homologs of the viral inserts, SA and SB, as antigens.

On day 28 after administering two rounds of the oral live recombinant probiotic vaccine Vac1 and Vac2 to Syrian golden hamsters, a low but statistically significant level of virus-neutralizing antibodies was detected through the plaque reduction neutralization test on Vero cell culture. The antibody titer was found to be 2–4 (Table 5), whereas no virus-specific antibodies were observed in the control group of animals that received oral administration of *E. faecium* L3.

The evaluation of the virus-neutralizing activity of the hamsters’ sera immunized with the oral live recombinant probiotic vaccines Vac1 and Vac2 using an enzyme-linked immunosorbent assay with human angiotensin-converting enzyme (ACE2) as the target showed that most of the sera had positive virus-neutralizing activity (Table 6). The mean values of virus-neutralizing activity were higher in the Vac1 and Vac2 groups than in the *E. faecium* L3 control group, although the differences were not statistically significant (*p* = 0.22 and *p* = 0.14, respectively). However, the virus-neutralizing activity of the sera in the Vac1 and Vac2 groups was statistically significant (*p* ≤ 0.05) compared to the untreated control group. Additionally, it should be noted that the average values of virus-neutralizing activity in hamsters treated with the original *E. faecium* L3 exceeded those in the untreated control group, but the difference was not statistically significant (*p* = 0.09).

Simultaneously, collected nasopharyngeal swabs of hamsters were analyzed using the same assay. The results showed that virus-neutralizing activity in swabs from hamsters treated with any of the probiotics was significantly higher compared to the untreated control group. The virus-neutralizing activity of swabs from hamsters in the Vac1, Vac2, and *E. faecium* L3 groups showed correlation with the level of virus-neutralizing activity in their blood sera, with correlation coefficients of 0.94, 0.49, and 0.92, respectively, suggesting a positive relationship between the two variables. In contrast, the correlation in the untreated control group was close to zero (−0.03).

An enzyme-linked immunosorbent assay (ELISA) was conducted to analyze specific IgG antibodies in the blood sera, using the full-length commercial SARS-CoV-2 S protein as the antigen. The results showed the presence of S-specific antibodies in the blood sera of hamsters orally vaccinated with the probiotics (Figure 3).

Additionally, blood sera from hamsters in the same group were analyzed using an ELISA assay with recombinant SA and SB proteins as antigens (Figure 4). The results showed the presence of specific IgG antibodies in the blood sera of hamsters immunized with the live vaccine.

### 3.2. The Protective Efficacy of Vac1 and Vac2 Oral Vaccines against SARS-CoV-2 Infection

Hamsters from all examined groups were orally presented with a dose of 4.3 lg PFU of the virus in a volume of 200 µL. Virus titer in the lungs of hamsters was quantified on days 3 and 6 post-infection. On day 6 post-infection, the SARS-CoV-2 virus load in the lungs of hamsters orally vaccinated with Vac1 and Vac2 strains showed a reduction of 82.07% and 99.56%, respectively, as compared to the control group receiving oral administration of *E. faecium* L3 (Figure 5 and Appendix A).

### 3.3. A Comparative Macroscopic and Histological Analysis of the Lungs during Coronavirus Infection

A macroscopic visual comparison of pulmonary pathology in Syrian hamsters following oral infection with SARS-CoV-2 was performed. Pathological changes in the lungs caused by the virus were divided into 5 groups and assessed on a scale from 0 to 4 depending on the degree of damage (Appendix A).

The lungs of untreated hamsters had normal anatomical and physiological characteristics, a pale pink color, and an unexpressed vascular pattern. The lungs were normal in volume and consistency, with smooth margins.

On the 3rd day after infection, when the accumulation of the virus in the lung tissue reached its peak, examination of the lungs of infected Syrian golden hamsters revealed almost the same lesions in all groups (*n* = 3/group). The lungs are plethoric, the vessels in the bronchial region are dilated. There were areas of both normal and focal inflammatory changes. The volume and consistency of the lungs are normal.

On the day 6 after infection, foci in the lungs continued to be detected in all experimental animals (*n* = 3/group). In the untreated control group, small (2–3 mm) hemorrhagic foci and medium focal pneumonia were found in the lungs. Compared to the untreated group, vaccination with both Vac1 and Vac2 resulted in less severe lung damage. Vac1-vaccinated hamsters showed lower lung involvement with small (approximately 1 mm) hemorrhagic lesions. The hamsters that received the Vac2 vaccine had the best lung condition. Slight inflammatory changes were observed with a grey-pink lung color similar to that of the *E. faecium* L3 strain control group.

Thus, post-mortem studies revealed the most severe forms of lung disease in untreated Syrian hamsters on the day 6 after infection with 4.3 log10 PFU/hamster SARS-CoV-2. The pathological anatomy of the lesion was characterized mainly by the development of mid-focal pneumonia. Figure 6f shows the results of lung damage assessment according to Appendix A.

According to microscopic examination of the lungs (Figure 6a–e) of untreated control animals, and those treated orally with *E. faecium* L3, Vac1, and Vac2, diffuse alterations in the alveoli were noted three days post-infection without the development of pneumonia foci. No intergroup differences were observed. Six days after infection, the lungs of animals displayed alternating areas of emphysema and atelectasis, and pneumonia foci with fibrinous-hemorrhagic exudate, which is characteristic of viral pneumonia. Based on the evaluation of lung injury markers, the Vac2 group exhibited the highest level of resistance to infection, whereas the untreated control group demonstrated the lowest resistance.

## 4. Discussion

Despite the successful development and licensing of several effective COVID-19 vaccines [43,44,45], widespread global vaccine deployment has not yet led to the achievement of herd immunity. Therefore, there is a need for ongoing development of new vaccine platforms that are cost-effective, readily available, and efficient.

The COVID-19 pandemic has accelerated the introduction of novel techniques for producing effective vaccines. This advancement may facilitate the emergence of new and diverse vaccines in the future, as traditional vaccine development methods have historically required a decade or more [46,47,48].

In addition to the established inactivated and subunit protein vaccines [49,50], novel technology-based approaches to combat the spread of COVID-19 have led to the development of viral vector vaccines [50] and mRNA vaccines [51,52].

The exploration of various methods to introduce vaccine antigens into the body will persist in the future to enhance vaccine safety, streamline antigen preparation procedures, and augment the immunization efficacy.

The potential and effectiveness of delivering vaccine antigens through live recombinant vaccines using probiotic microbial strains have been validated by numerous laboratory studies [1,2,39,53]. A number of these live vaccines are in various stages of clinical trials [54,55].

The study of two live recombinant probiotic vaccines, Vac1 and Vac2, was conducted to evaluate their immunogenic and protective properties. The genomes of these vaccines were modified by insertion mutagenesis, resulting in the insertion of DNA fragments encoding partially overlapping sequences of the coronavirus S1 protein. This method for constructing Vac1 was previously described by us [40]. Analysis of the vaccine strain showed expression of the viral antigen on the surface of the bacterial cells, as demonstrated by immune electron microscopy using serum from SARS-CoV-2 patients.

Vac2 was constructed similarly to Vac1 and both vaccine strains were based on *E. faecium* L3. The inserted DNA fragments encoding different regions of the S1 protein of the coronavirus were of similar size and located in the same region of the gene encoding the main pili protein, which is exposed on the surface of bacteria [40].

A fragment of the S1 protein of coronavirus, including a partial region of RBD (496–646, GenBank: OL447006.1), was inserted into the genome of Vac1. The insert contained a significant portion of B- and T-dependent determinants, in addition to the RBD region. It has been established that the evolution of pandemic strains occurs mainly due to mutations in the RBD region [56]. Thus, the choice of this particular antigenic region of the coronavirus spike protein suggests that the vaccine’s effectiveness should not be significantly impacted by the process of antigenic variability of the pathogen under immune selection pressure.

The Vac2 vaccine strain was constructed with a coronavirus DNA insert encoding the full-length receptor-binding domain (RBD) of the S1 protein (positions 364–533, GenBank: ON803610.1). The original DNA sequence of SARS-CoV-2 was modified to incorporate changes in key residues responsible for ACE2 binding, including K417N, E484K, and N501Y. The resultant amino acid sequence corresponded to the beta coronavirus lineage B.1.351 (SARS-CoV-2), generating antibodies that can bind not only the Beta but also Gamma and Omicron lineages of coronaviruses [57]. The fragment contained both MHC class I and MHC class II restricted antigenic determinants, according to the Immune Epitope Database.

Both strains of the live recombinant probiotic vaccine were tested in hamsters. They expressed different fragments of the coronavirus S1 protein, with one of them containing the entire RBD domain. The live vaccines were administered orally, thus eliciting an immune response through the mucous membranes of the oral cavity and digestive system. Animals treated orally with the original *E. faecium* L3 variant served as controls, receiving the same dose and following the same schedule (Figure 1).

Observation of hamsters during mucosal and parenteral vaccination did not reveal any significant differences in terms of body weight gain, behavior, or physiological parameters (Table 4), indicating the safety of the vaccination procedures.

The immune response to mucosal vaccination was evaluated through several assays. Two of them measured the level of virus-neutralizing antibodies in the blood serum of the experimental animals (Table 5 and Table 6). A conventional ELISA was also conducted, where commercial full-length S protein or recombinant analogues (SA and SB) of the coronavirus inserts were adsorbed onto plates as antigens (Figure 3 and Figure 4). According to ELISA data conducted with a commercial protein (Figure 3), the immune response parameters for Vac1 and Vac2 were significantly different from the control, as determined by nonparametric analysis.

The results of all tests showed that the level of circulating virus-specific antibodies after oral vaccination with both vaccine strains was moderate. Thus, the analysis of the neutralizing activity of the serum from vaccinated hamsters revealed a reciprocal titer of 2–4 in the neutralization reaction on Vero cells (Table 5). At the same time, no neutralizing activity was observed in the control group. Thus, oral vaccination with both Vac1 and Vac2 led to the accumulation of factors in the blood serum that could bind to the SARS-CoV-2 S protein and prevent its entry into Vero cells. The absence of a significant difference in the neutralization value between the Vac1 and Vac2 groups might have been due to the low level of humoral specific immune response.

Further analysis of the serum by enzyme immunoassay revealed that most of the blood serum from hamsters immunized with Vac1 and Vac2 had a positive virus-neutralizing activity against the human angiotensin-converting enzyme (ACE 2) (Table 6). This suggests that the serum contained factors that could bind the coronavirus S protein in solution and prevent its interaction with ACE 2.

In this study, we evaluated the virus-neutralizing properties of swabs collected from the oral mucosa. Regrettably, our results indicate that the neutralization index of all groups analyzed was below the established positive threshold of 20% for this method (Table 6). This finding may be attributed to the lower sensitivity of antibody detection procedures in secretions as compared to serum, as well as to the lower concentration of specific IgG antibodies in saliva [58,59].

It is noteworthy that the S protein-binding efficacy in the buccal swabs of hamsters treated with probiotics in the forms of Vac1, Vac2, and Enterococcus faecium L3 (Table 6) was significantly higher than that of untreated hamsters. This effect can be elucidated by the immunomodulatory properties of probiotic bacteria, which have been extensively described in the scientific literature [60,61,62]. Probiotics have demonstrated their capability to stimulate the innate immune system, which involves the production of natural antibodies that exhibit broad-spectrum antimicrobial and antiviral activities [63,64].

We examined the correlation between the virus-neutralizing activity levels in sera and buccal swabs of vaccinated animals. Although a positive correlation was observed, statistical significance was not achieved, most likely due to the limited sample size in the experimental groups. In the literature, there are varying opinions on the relationship between serum antibody levels and secretions following infection or vaccination. Several studies support a positive correlation between specific IgG and IgM, but a weaker correlation between IgA.

An ELISA was conducted on blood sera to detect virus-specific IgG using a commercial full-length SARS-CoV-2 S protein (Figure 3) and recombinant SA and SB polypeptides that were homologous to the viral inserts in Vac1 and Vac2 (Figure 4). The results indicate that the blood sera of hamsters in both Vac1 and Vac2 groups contained S-specific antibodies that could interact with the commercial full-length S protein. Statistically significant differences were observed between the Vac1 group and the control group, while the differences between the Vac2 group and the control group were slightly higher than 0.05, which might be improved with a larger sample size of animals (Figure 3).

The pooled serum samples from each group were subjected to analysis using recombinant proteins SA and SB as antigens in ELISA (Figure 4). While the pool of immune sera exhibited elevated OD450 values compared to the control animals, the absence of individual analyses precluded any statistical data processing and only permitted the observation of a trend.

Therefore, our study demonstrated that oral vaccination with probiotic recombinant strains carrying two different fragments of the S1 coronavirus protein elicits a specific humoral immune response to viral proteins.

From the perspective of conventional vaccinology, a moderate humoral immune reaction to a vaccine is deemed unfavorable. Nonetheless, findings from investigations into mucosal vaccines suggest that protection against infection can be attained even with low quantities of specific antibodies present in the bloodstream [1,63,65].

Despite a moderate humoral immune response from oral vaccination, the hamsters were protected from oral coronavirus infection. Vac2, which carried the DNA sequence encoding the complete RBD motif of spice protein with substitutions in positions E484K, N501Y, exhibited the highest level of protection.

For the infection, the SARS-CoV-2 virus strain hCoV-19/Russia/SAB-1502/2021, which carries amino acid substitutions in the spike protein D215G, E484K, N501Y, was selected. As previously mentioned, despite being similar to the beta coronavirus variant, the amino acid sequence of Vac2 was found to stimulate antibodies that neutralize not only beta but also gamma and omicron coronavirus strains, according to previous reports [57].

The viral insert in the Vac1 vaccine was constructed based on the GenBank sequence OL447006.1, which corresponds to the earliest Wuhan version of SARS-CoV-2 and precedes the antigenic group of gamma viruses. Vac1 contained only a portion of the RBD, while the majority of the coronavirus insert did not belong to RBD and is reportedly less susceptible to mutation according to the literature [57]. As a result, the coronavirus protein in Vac1 and the virus used for infection may share several antigenic epitopes.

The oral route of infection was chosen due to the several reasons. Coronavirus spreads through various transmission pathways in real-world situations, including oral transmission through close interpersonal contact or exposure to exudates from infected individuals during coughing or sneezing episodes. It is well established that coronavirus can penetrate the human body through multiple means, not limited to the susceptible cells of the upper respiratory tract. Studies have shown that the epithelium of the oral cavity [66] and the gastrointestinal tract [67] can also support virus replication.

Our selection of the infection method took into account that the strongest immune response would occur at the mucous membrane where the vaccine was administered. This led us to expect that the oral vaccine’s protective effectiveness would be highest when combined with the oral infection route.

Using the oral infection method offers better precision and standardization for introducing the virus in laboratory settings compared to aerosol-based methods. This meticulous control ensures consistent viral dosing for all experimental subjects. Moreover, prioritizing participant safety makes oral infection a preferable and safer choice.

Therefore, the choice of infection method aimed to establish optimal experimental conditions while aligning with documented modes of infection transmission during the epidemic. It is important to note that we have previously demonstrated the efficacy of probiotic oral vaccines against viral and bacterial respiratory infections too [1,53]. This suggests a potential similar outcome for coronavirus infection, but empirical validation is necessary.

Oral administration of the virus resulted in a full-fledged coronavirus infection and virus entry into the hamsters’ lungs. The development of the infection was evaluated through anatomical and histological examination of the lungs on days 3 and 6 post-infection, as well as through measurement of the viral load in the lung at the same time.

Administration of recombinant Vac1 and Vac2 vaccine strains reduced the accumulation of the virus in the lungs. The results from Figure 5 showed that on day 6 post-infection, the reduction was 82.07% and 99.56%, respectively, compared to the control group treated orally with *E. faecium* L3. The vaccine strain with the highest efficacy in suppressing viral replication in lung tissue was Vac2, which contained the RBD and was antigenically similar to the infecting virus. Vaccination with Vac1 also demonstrated a significant protective effect.

Despite the moderate level of humoral immune response, live recombinant probiotic vaccination effectively suppressed coronavirus infection. Other mechanisms, such as cellular cytotoxicity, may have also played a role in the antiviral protection. The literature suggests that mucosal vaccination with probiotic bacteria carrying a pathogenic antigen can induce such specific mechanisms [68,69].

The presence of protection despite a relatively weak humoral immune response is of interest in the context of the discussion on the consequences of an inadequate immune response during infection, as excessive, inappropriate, or altered immune responses can lead to increased tissue damage, impaired lung function, and worsening of respiratory disease [70].

Pathological and histological comparison of lung damage during infection showed that coronavirus caused the most severe damage in untreated hamsters (Figure 6). The hamsters vaccinated with Vac2 were least affected by the infection.

The data suggests that the recombinant vaccine strains created from the probiotic strain *E. faecium* L3, which express immunogenic fragments of the S1 protein of coronavirus, can effectively suppress SARS-CoV-2 infection in laboratory hamsters after oral vaccination. The vaccine strains, both containing RBD and encoding other antigenic domains of the S1 protein, limit viral reproduction. The technology for producing the recombinant vaccine strains is simple and easily reproducible, making it possible to quickly develop new vaccines to adapt to changes in the pathogen’s antigenic properties under the pressure of the immune response. Probiotic-based vaccines, such as the studied strains or similar ones, may have advantages in terms of safety, ease of production and use, and, therefore, low cost and accessibility.

## 5. Conclusions

In summary, the data presented here have demonstrated that the employed method for integrating the DNA sequence of coronavirus protein S into the genome of probiotic strain *E. faecium* L3 enables the generation of vaccine strains. These strains, when administered orally, induce a protective immune response in hamsters against coronavirus infection.

Such vaccines, characterized by their simplicity and reproducibility in production, offer a prompt response to the emergence of novel antigenic virus variants. Probiotic-based vaccines, akin to those investigated, exhibit potential advantages concerning safety, cost-efficiency, and accessibility.

## Figures and Tables

**Figure 1 vaccines-11-01714-f001:**
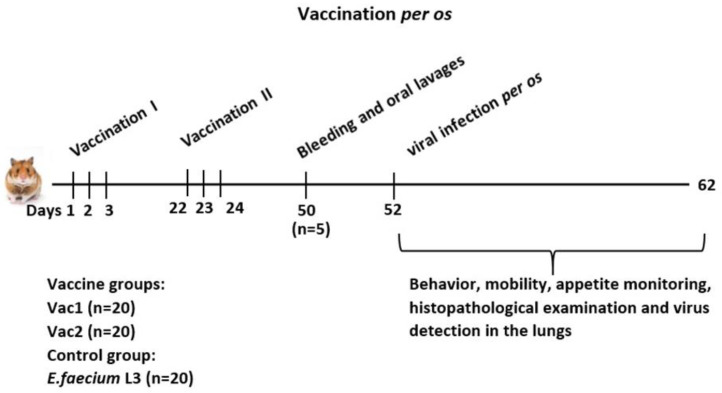
Experimental setup.

**Figure 2 vaccines-11-01714-f002:**
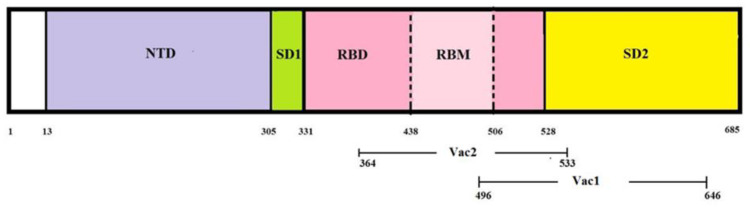
The structure of the coronavirus inserts in a live probiotic vaccine.

**Figure 3 vaccines-11-01714-f003:**
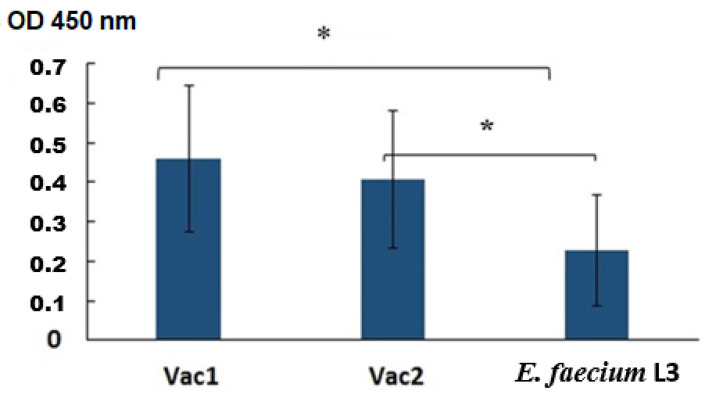
Analysis of specific IgG in sera by ELISA. The OD450 nm values of sera diluted 1:100 are shown. The results are presented as the mean ± SEM (*n* = 5/group). *—A statistically significant difference between the two groups was identified using a Mann–Whitney U test.

**Figure 4 vaccines-11-01714-f004:**
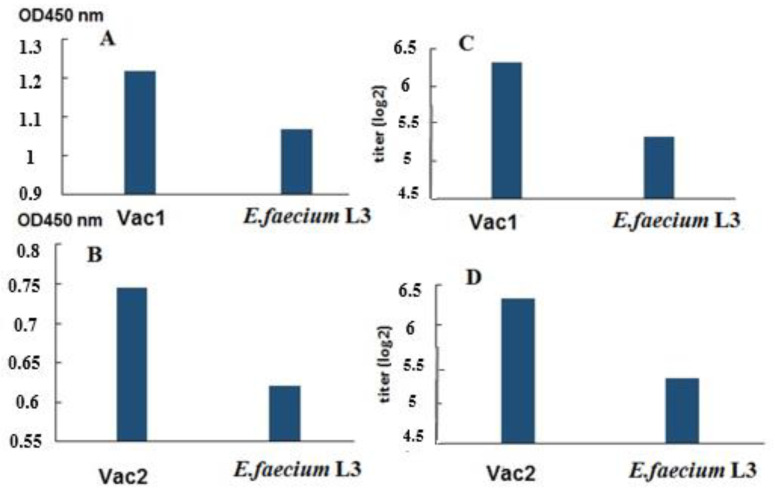
Analysis of specific IgG in pooled blood sera in ELISA with SA and SB proteins as antigens (*n* = 5/group). After administering the Vac1 (**A**,**C**) and Vac2 (**B**,**D**) vaccines, blood sera from individual animals were combined and analyzed using an ELISA assay to assess the presence of IgG antibodies against recombinant SA and SB proteins, which are similar to the coronavirus components in Vac1 and Vac2, respectively. (**A**,**B**) Comparison of sera in ELISA based on OD450 values. The optical density at 450 nm (OD_450_) was determined for serum diluted 1:20. (**C**,**D**) Comparison of sera in ELISA based on the end-point ELISA titers. To determine the titer, the serum dilution was considered at which the OD450 was five times higher than that of the serum from an untreated hamster (OD = 0.240).

**Figure 5 vaccines-11-01714-f005:**
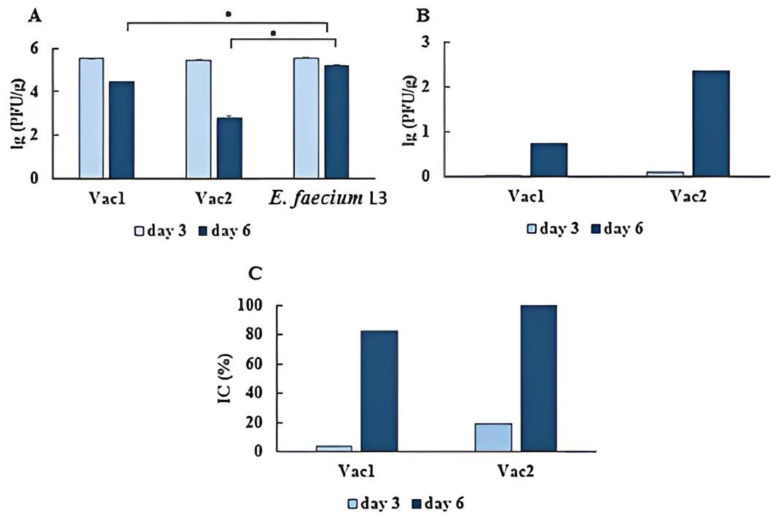
Analysis of SARS-CoV-2 virus load in the lungs on days 3 and 6 after oral vaccination. Syrian hamsters were orally administered Vac1, Vac2, and *E. faecium* L3 according to the schedule in Materials and Methods. Hamsters were orally challenged with SARS-CoV-2 28 days after the second vaccination course. After 3 (*n* = 3/group) and 6 (*n* = 3/group) days from the onset of infection, the lungs were obtained. The viral load (**A**), decrease in viral load (**B**), and inhibition coefficient (**C**) were determined. •—A statistically significant difference between the two groups was identified using a Mann–Whitney U test.

**Figure 6 vaccines-11-01714-f006:**
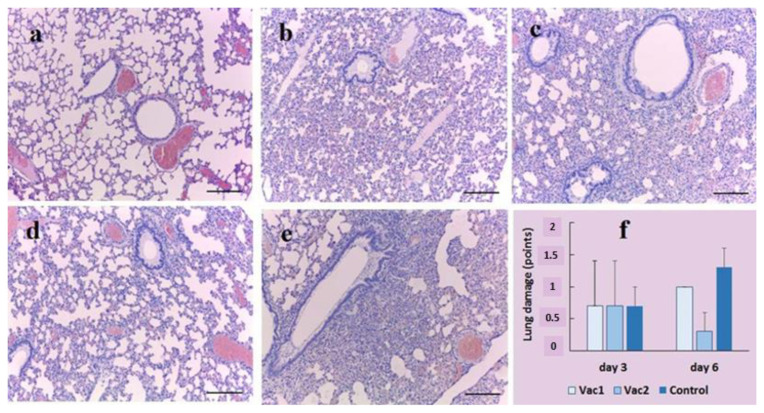
Histological and macroscopic analysis of lung tissue. The Syrian hamster lungs were examined on day 6 using Hematoxylin and eosin stain: untreated and uninfected control animals (**a**); animals that were orally vaccinated with *E. faecium* L3 (**b**); live recombinant probiotic vaccines Vac1 (**c**) and Vac2 (**d**); untreated and infected hamsters (**e**) (*n* = 3/group). The pulmonary pathology in Syrian hamsters after SARS-CoV-2 virus infection on day 3 and 6 post-SARS-CoV-2 virus infection (*n* = 3/group/point) was quantified using a scoring system (**f**). Hematoxylin and eosin staining x10. Scale bar corresponds to 200 μm.

**Table 1 vaccines-11-01714-t001:** Characteristics of an infectious virus.

Virus Strain	Passages	Concentration (lg PFU/mL)	Concentration (TCID50/mL)
SARS-CoV-2, hCoV-19/Russia/SAB-1502/2021 strain	Two passages in Vero Cl008 cell	6.4	5.5

**Table 2 vaccines-11-01714-t002:** SARS-CoV2 sequences incorporated into *E. faecium* L3 chromosome.

*E. faecium* L3 with SARS-CoV2 Insertion	Inserted Sequence (GenBanc Link)	Recombinant Polypeptide Analogue of Insert
Vac1	Vac1 Synthetic construct of partial S1 spike protein gene, cds. GenBank: OL447006.1 https://www.ncbi.nlm.nih.gov/nuccore/OL447006.1/, accessed on 20 January 2020	SA
Vac2	Synthetic construct of partial S1 spike protein gene, cds. GenBank: ON803610.1	SB

**Table 3 vaccines-11-01714-t003:** Expression of SA and SB proteins in *E. coli*.

Recombinant Protein Name	Vector	Producer Strain	Primers	Direction 5′ to 3′	Nucleotide Sequence from 5′ to 3′	Purpose
SA	pQE-30 (Qiagen, Hilden, German)	*E. coli* M15	Cov1	F	AAGGATCCATACATATGGGTTTCC	Cloning a gene fragment for protein SA production
			Cov2	R	TGTCGACGGAGCTCGAATT	Cloning a gene fragment for protein SA production
SB	pET-22b (Qiagen, Hilden, German)	*E. coli* BL21	CS1	F	TTGCATATGGATTATTCTGTCCTATATA	Cloning a gene fragment for protein SB production
			Cv22	R	CCAAGCTTAGTAGACTTTTTAGGTCCACA	Cloning a gene fragment for protein SB production

**Table 4 vaccines-11-01714-t004:** Effect of vaccination on body weight of Syrian golden hamsters.

Experimental Groups	Body Weight of Animals in the Group (g) ± SEM								
	Day 0	5	10	15	20	25	30	35	40
VacA	54.9 ± 0.5	60.1 ± 0.8	66.3 ± 0.2	73.6 ± 1.8	77.3 ± 1.6	86.5 ± 1.1	89.8 ± 2.6	91.2 ± 3.3	94.3 ± 1.1
VacB	51.2 ± 1.8	53.4 ± 1.6	59.5 ± 1.2	65.1 ± 1.2	69.0 ± 0.9	78.2 ± 2.5	82.0 ± 3.4	84.9 ± 1.4	89.0 ± 2.1
*E. faecium* L3	54.3 ± 0.6	58.1 ± 1.5	63.1 ± 1.6	66.7 ± 1.3	70.9 ± 1.5	76.4 ± 1.7	79.2 ± 2.2	83.4 ± 1.1	87.5 ± 1.0

**Table 5 vaccines-11-01714-t005:** The level of virus-neutralizing antibodies in Syrian golden hamsters in the Plaque reduction neutralization test on Vero cells.

Experimental Groups	Number of Animals	Reciprocal Titers
Vac1	5	(2–4)
Vac2	5	(2–4)
*E. faecium* L3	5	˂2

**Table 6 vaccines-11-01714-t006:** Evaluation of the level of virus-neutralizing activity in Syrian golden hamsters by inhibition of S-protein binding to human ACE 2 in ELISA.

Experimental Groups	Sample Number *	The Neutralization Index of Serum (%) ** m ± SEM		The Neutralization Index of Swabs (%) ** m ± SEM	
Vac1	1	25.2	22.4 ± 7.7	14.5	13.1 ± 3.9
	2	37.5		19.7	
	3	23.8		14.8	
	4	4.8		6.8	
	5	20.7		9.6	
Vac2	6	15.0	26.6 ± 12.0	5.8	11.6 ± 3.0
	7	43.2		11.7	
	8	38.6		12.9	
	9	8.2		9.7	
	10	28.1		17.7	
*E. faecium* L3	11	17.5	14.1 ± 6.2	21.4	17.5 ± 5.2
	12	11.9		14.5	
	13	26.1		26.7	
	14	7.9		16.2	
	15	7.1		8.7	
Untreated control group	21	1.8	5.9 ± 2.1	5.5	5.3 ± 1.7

* Each sample number corresponds to the results obtained from the study of serum and swabs collected from the same animal. ** According to the method, a neutralization index level of more than 20% was taken as a positive value.

## Data Availability

All data are contained in the text of the article and Appendix A files.

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
