# Peer review of "SARS-CoV-2 Spike Protein-Expressing Enterococcus for Oral Vaccination: Immunogenicity and Protection"

_vaccines, 2023, doi:10.3390/vaccines11111714_

Round 1

Reviewer 1 Report

Comments and Suggestions for Authors

In this article Suvorov et al highlight the concerns relating to the COVID-19 pandemic and the continuing appearance of new mutations and periodic surges in the number of cases of coronavirus infections. They comment on the fact that current vaccination strategies may not continue to be effective over time and they highlight the issues regarding continual virus mutations that make it difficult to keep ahead of the ever-changing landscape for vaccine intervention.

In this study the authors have developed a very convenient and highly universal technique for inserting the DNA sequences of pathogenic bacteria and viruses into the gene that encodes the pili protein of the probiotic strain E. faecium L3. The authors wanted to assess and investigate the protective efficacy of two recombinant enterococcal strains designated as Vac1 and Vac2 as probiotic vaccines. The same approach was used for the construction of Vac1 and Vac2.  A 533 bp S1 DNA fragment encoding the complete receptor-binding domain of the SARS-CoV-2 virus was selected from the database and this was used for insertion into the probiotic genome. The authors then generated purified recombinant proteins homologous to the insert in order to analyse the specific immune response to the viral polypeptides expressed by the probiotic bacterium. This manuscript presents data on the immunogenic properties of two E. faecium L3 vaccine strains, which produce two different fragments of the coronavirus S1 protein, and provides an assessment of the protective efficacy of these oral vaccines against coronavirus infection in Syrian hamsters.

This is a hugely important area of research and obviously a very strategic scientific endeavour to follow through as the cases of coronavirus are again increasing as we head towards the winter months and there is a real need for a different approach to be taken in order to combat the high level of mutation found in these viruses. The creation of safe vaccine platforms that can be rapidly adapted to deliver new, specific antigens in response to viral mutations will be a huge benefit to society. The authors are careful to explain that even though existing vaccines are undeniably effective at preventing coronavirus, it is important to work towards improving any existing vaccine platforms and also to develop safer vaccine candidates if possible. As the authors state in their manuscript, the use of recombinant probiotic bacterial strains to deliver vaccine antigens through the gastrointestinal mucosa could be a safe and effective vaccine strategy. Based on the results they obtained, it would appear that oral administration of the strains resulted in an immune response that was effective in protecting against a SARS-CoV-2 infection. The authors state that the inhibition of virus reproduction was 82% and 99% in the vaccinated hamsters compared with the control group. This was also supported by the data from the morphological analyses of the lungs. Based on these data, the authors suggest that immunisation with probiotic vaccine candidates could serve as a potential alternative to injectable vaccines against coronaviruses.

Main points and comments:

1.     The experimental design and strategy appears to be suitable and well controlled. The rationale for the experiments is robust and clearly defined.

2.     The Supplementary files are useful and help to explain and underpin some of the data that was generated.

3.     The authors stated that “The selection of the virus was grounded in the observation that, during that specific period, both the previous Beta variant and the emerging Gamma variant of the coronavirus were concurrently in circulation.” Have the authors considered assessing any other variants to compare the data they have in this manuscript?  There are now a large number of variants that the authors could consider for analysis.

4.     Can the authors comment on how close they think we are to obtaining herd immunity?

5.     The authors state that the Syrian golden hamsters they used in the experiments were maintained in plastic cages with the temperature kept between 15 and 21 ˚C and the humidity at between 30-70%, with a 12-hour light/12-hour darkness cycle. The temperature and humidity ranges appear to be quite large. Is this the normal situation for experimental animals in this facility? Did the hamsters have any environmental enrichment and if so, what? What was the stocking density for the experimental animals?

6.     The authors report that they used an oral dose of 200μl of SARS-CoV-2 virus at a dose of 4.3 log PFU for each hamster. Was this the only dose they tried? Why did they decide on this dose of SARS-CoV-2?

7.     Section 2.12 Line 297. Can the authors please list exactly which features were documented during the visual inspection as the use of the word “etc” is neither scientific nor helpful and is certainly not giving clarity to the reader.

8.     Can the authors comment on how effective an oral coronavirus vaccination is likely to be in other species, not just the Syrian hamsters.

9.     Do the authors have any data relating to post-vaccination complications in their hamster model?

10.  Are side-effects from an oral vaccination less prevalent in the hamster model as opposed to other animal models?

11.  How robust do the authors feel their model is and would they be confident of the outcome if different coronaviruses were used to test the system they have employed?

12.  Table 2 would benefit from not being placed over 2 pages.

13.  All the Tables would benefit from a bit of a tidy up regarding column headings and neatness.

14.  Please can the authors be consistent with their labelling of items such as Figures and Tables. For example line 161 states Figure 1; line 255 (Fig.S1-S3); line 261 (Fig. 2); line 300 (Supplement Table S1); line 340 (Fig. S4); line 429 (Table S1) – Is this the same Supplementary Table as is listed on line 300? There are a variety of spacings and full stops and wording that would benefit from being consistent.

15.  Table 4, I think the word ‘Day’ should be placed in front of the 0 in the first data column rather than in front of the ‘5’ in the second data column. This table should also state ‘± SD’ or '±SEM' in the heading to account for all the standard deviations or standard error of mean that are shown in the data sets.

16.  Table 6. What are the units used in the Table? There appear to be 2 columns not labelled. Are these means ± standard deviations? Please label clearly.

17.  Figure 4. Is it possible to carry out some statistical analyses for this Figure as for Figure 3?

18.  Figures S2 and S3 in the supplementary file appear to be Tables. Can these be labelled correctly, and the adjusted information added to the text where needed?

19.  In the Supplementary data file, it is quite difficult to follow as some headings are above the Tables / Figures and some headings are below them. Can these be made consistent please?

20.  Are there some statistical analyses missing from Figure 5B and 5C?

21.  Supplementary files – Table S1 – What are the units for time after infection (3 or 6 what)? Table S3 – please spell out the (m±SEM) to make it clear.

22.  There are a few typographical errors and the odd random full stop in a strange place (such as on page 5 of the Supplementary data where the sentence starts ‘.Among Europeans, the most common………’. Please be consistent throughout.

23.  Can the OD450nm please be added to the relevant Figures rather than just ‘OD450’.

This is an interesting paper, and the data support the conclusions that have been suggested. The study will need to be expanded on but this is a good start to finding alternative vaccination strategies for coronaviruses other than injectable vaccines (at least in the hamster).

Comments on the Quality of English Language

This is a reasonably well written manuscript, and the use of English language is good. There are just a few minor typographical errors that would benefit from being corrected and some tidying up needed regarding the presentation of data and the labelling of Tables and Figures.

Author Response

The authors express their deep gratitude to the Reviewer  for the meticulous analysis of our manuscript. Below, we address his comments that require a response and delineate the corresponding textual modifications that align with the essence of these comments.

  1. The authors stated that “The selection of the virus was grounded in the observation that, during that specific period, both the previous Beta variant and the emerging Gamma variant of the coronavirus were concurrently in circulation.” Have the authors considered assessing any other variants to compare the data they have in this manuscript?  There are now a large number of variants that the authors could consider for analysis.

The research was conducted during a period of concurrent circulation of beta and gamma coronaviruses. The selected viral strain for hamster infection exhibited structural homology with both of these genetic variants. Following the inoculation of immunized hamsters, it became evident that both vaccine formulations expedited viral clearance from pulmonary tissues. The Vac 2 vaccine variant, encompassing the complete Receptor Binding Domain (RBD) of the coronavirus S protein, demonstrated the highest level of protective efficacy, achieving a remarkable 99.6% effectiveness. In contrast, the Vac1 vaccine lacked a full-length RBD insert; however, it still reduced viral replication in the lungs by 82.07%. It is apparent that a close structural correspondence between the vaccine antigen and the viral pathogen represents an optimal configuration. This observation, though speculative, suggests that mucosal vaccination employing the investigated live recombinant vaccines may potentially prove effective against emerging coronavirus variants harboring mutations within the RBD region of the S protein.

 In addition to in vivo experiments, which offer compelling evidence of protective efficacy, alternative approaches exist for in vitro assessment of serum neutralization against novel coronavirus S protein variants. The hamster sera obtained during the experiments described in this publication were utilized for this purpose, precluding the feasibility of conducting further retrospective investigations.

 The development of live probiotic recombinant vaccines tailored to address new SARS-CoV-2 variants remains an ongoing endeavor.

 This year, Eddie Chung Ting Chau et al. published a study detailing the construction of a live recombinant vaccine utilizing Lactobacillus casei as the expression host for the SARS-CoV-2 Omicron variant B.1.1.529. This was achieved through the cloning of the corresponding DNA fragments into the highly efficient expression plasmid pPCT2. The authors demonstrated that this vaccine elicited a robust systemic IgG immune response targeting the spike protein of Omicron variant B.1.1.529 in Golden Syrian hamsters. However, it is essential to note that the study did not investigate the vaccine's protective effectiveness against the Omicron strain or other contemporary coronavirus variants.

  1. Can the authors comment on how close they think we are to obtaining herd immunity?

We acknowledge that our insights in the field of epidemiology are not those of experts, and it is possible that specialists may not fully endorse the considerations articulated below.

The concept of herd immunity hinges on the establishment of a substantial immune population within society, which acts as a barrier to the pathogen's transmission. For this to materialize, several prerequisites must be met:

-Immune individuals must possess enduring and stable immunity.

-This immunity should effectively neutralize the virus and inhibit its replication.

-The virus should cease to propagate upon encountering immune individuals.

In this context, we currently appear distant from achieving herd immunity. The longevity of immunity to the coronavirus is limited for several reasons. Repeated infections with the coronavirus, occurring within relatively short intervals – spanning one year, two years, or even less – underscore that both post-vaccination and natural immunity fail to provide complete sterilization even over a one-year period.

On another note, there is evidence that over time, the course of coronavirus infections tends to be milder. This phenomenon may be attributed to the development of immunological memory against the entire spectrum of antigenic determinants of the coronavirus, including conserved elements. While this process does not eliminate infections, it does facilitate their course, gradually shifting the dire pandemic of the coronavirus toward the realm of seasonal challenges.

It is plausible that attaining collective immunity against the coronavirus might prove elusive. Instead, a continued effort akin to that employed in preventing influenza will likely be necessary. In such a scenario, mucosal vaccines built on recombinant probiotics may emerge as pivotal tools in this ongoing battle.

  1. 5. The authors state that the Syrian golden hamsters they used in the experiments were maintained in plastic cages with the temperature kept between 15 and 21 ˚C and the humidity at between 30-70%, with a 12-hour light/12-hour darkness cycle. The temperature and humidity ranges appear to be quite large. Is this the normal situation for experimental animals in this facility? Did the hamsters have any environmental enrichment and if so, what? What was the stocking density for the experimental animals?

The housing and care of the animals adhered to established international guidelines, specifically following the "Recommendation for the monitoring of mouse, rat, hamster, guinea pig, and rabbit colonies in breeding and experimental units" as outlined by the European Laboratory Animal Science Associations (FELASA) in 2014. In addition, local standards, in accordance with SP No. 1045 73 for the maintenance and arrangement of vivariums (GOST R 53434-2009), were strictly observed. Furthermore, the "Guide to laboratory animals and alternative models in biomedical technologies," published in Moscow in 2010, was consulted for reference.

The parameters governing the animals' environment were reported in the Materials and Methods section as is customary in regulatory documents. The experimental conditions maintained an average temperature of 20°C and humidity levels at 60%, with no supplementary environmental enrichment measures implemented.

The stocking density for the animals was maintained at a minimum of 100 cm2 per individual.

  1. The authors report that they used an oral dose of 200μl of SARS-CoV-2 virus at a dose of 4.3 log PFU for each hamster. Was this the only dose they tried? Why did they decide on this dose of SARS-CoV-2?

A single dose was exclusively employed in this study, selected based on prior evidence demonstrating its efficacy in initiating coronavirus infection in hamsters. This particular dosage facilitated the establishment of a suitable model for evaluating the protective efficacy of vaccination.

  1. Section 2.12 Line 297. Can the authors please list exactly which features were documented during the visual inspection as the use of the word “etc” is neither scientific nor helpful and is certainly not giving clarity to the reader.

We have refined the manuscript's wording and made appropriate revisions.

  1. Can the authors comment on how effective an oral coronavirus vaccination is likely to be in other species, not just the Syrian hamsters.

The widely acknowledged experimental model for simulating coronavirus infections involves the use of Syrian golden hamsters and humanized mice. In our research, we exclusively utilized the Syrian golden hamster model and did not employ the latter. Additionally, we conducted our independent investigations concerning the immunogenic properties of live recombinant coronavirus vaccines in vaccinated mice from the CBA and Balb/c strains. Our findings demonstrated that oral vaccination elicited both local and systemic humoral immune responses and established immunological memory. According to our data, oral vaccination holds promise as a valuable component of a prime-boost vaccination approach. The results of these studies are currently prepared for publication and are in the process of undergoing peer review.

  1. Do the authors have any data relating to post-vaccination complications in their hamster model?

We assessed the safety of probiotic live vaccines through a comprehensive evaluation, which encompassed weight measurements and observations of various parameters such as behavior, body composition, hair condition, skin health, condition of mucous membranes, excrement characteristics, and respiratory patterns. All observations were consistently within the expected and normal physiological ranges

.10.  Are side-effects from an oral vaccination less prevalent in the hamster model as opposed to other animal models?

Our vaccines are formulated with E. faecium L3, a probiotic microorganism that lacks harmful pathogenic genes. This particular strain has a well-established history of serving as a beneficial probiotic supplement in human nutrition. Importantly, no adverse effects or side effects have been reported in our department with its utilization in experimental animals. In fact, E. faecium L3 has proven to be effective in our department's experiments involving rats, particularly in the treatment of dysbiotic conditions induced by antibiotic administration.

  1. How robust do the authors feel their model is and would they be confident of the outcome if different coronaviruses were used to test the system they have employed?

First and foremost, it is crucial to emphasize that any assumption must be subject to rigorous verification. In addressing the posed question, it is reasonable to consider that the comparative effectiveness of protection will be directly influenced by the level of antigenic similarity between the coronaviruses employed and the antigenic properties of coronavirus inserts incorporated into our probiotic vaccines. This principle holds true for all vaccine formulations.

Nonetheless, we posit that the mucosal route of administration and the utilization of beneficial probiotic microorganisms as carriers for vaccine antigens offer certain advantages in the context of our vaccines. Initially, it is well-established that applying an antigen to mucosal surfaces primarily triggers a specific immune response within the mucosal compartment, which is the entry point for pathogens. In contrast, parenteral vaccination primarily elicits a systemic immune response, with less pronounced mucosal reactions.

Furthermore, probiotic bacteria possess inherent capabilities to stimulate innate defense mechanisms, including those operating at mucosal surfaces. Consequently, it can be postulated that upon viral entry into the mucosal domain, it may undergo neutralization, leading to a reduction in its infectious potential. This effect is likely to be more pronounced when employing mucosal probiotic vaccines.

Therefore, we surmise that by employing various viruses in our model, we will be able to observe the protective effects of vaccination. However, predicting the exact extent of this protection remains a complex challenge.

  1. Table 2 would benefit from not being placed over 2 pages.

This issue will likely be corrected during final printing. We have omitted the table below in the draft manuscript, so it now spans one page.

  1. All the Tables would benefit from a bit of a tidy up regarding column headings and neatness.

The journal adheres to a format for tables that lacks dividing lines between columns. We kindly request you to organize the column headings in the tables accordingly.

  1. Please can the authors be consistent with their labelling of items such as Figures and Tables. For example line 161 states Figure 1; line 255 (Fig.S1-S3); line 261 (Fig. 2); line 300 (Supplement Table S1); line 340 (Fig. S4); line 429 (Table S1) – Is this the same Supplementary Table as is listed on line 300? There are a variety of spacings and full stops and wording that would benefit from being consistent.

In accordance with the stipulations outlined in section 14 and the feedback provided in section 18, revisions have been applied to the supplementary materials file and the main manuscript text.

  1. Table 4, I think the word ‘Day’ should be placed in front of the 0 in the first data column rather than in front of the ‘5’ in the second data column. This table should also state ‘± SD’ or '±SEM' in the heading to account for all the standard deviations or standard error of mean that are shown in the data sets.

Table 4 has been rectified as per the recommendations provided.

  1. Table 6. What are the units used in the Table? There appear to be 2 columns not labelled. Are these means ± standard deviations? Please label clearly.

Table 6 has been rectified as per the recommendations provided.

  1. 17. Figure 4. Is it possible to carry out some statistical analyses for this Figure as for Figure 3?

This figure visually depicts the outcomes of the analysis conducted on pooled sera, obtained by combining the sera from individual animals within each study group in equal proportions. Logically, the data acquired should reflect an estimation of the average immunogenicity within each group. However, it's important to note that these data are not amenable to statistical analysis and can only be regarded as supplementary indications of the immunogenicity associated with the vaccines under investigation. In the discussion section, we have included these data while explicitly mentioning their absence of statistical processing.

  1. Figures S2 and S3 in the supplementary file appear to be Tables. Can these be labelled correctly, and the adjusted information added to the text where needed?

The format of the supplementary file has been modified, and the necessary adjustments have been applied to the manuscript text.

  1. In the Supplementary data file, it is quite difficult to follow as some headings are above the Tables / Figures and some headings are below them. Can these be made consistent please?

We have revised the supplementary materials file as per  instructions. The table headings are now positioned above the tables, and the caption texts are situated below the figures.

  1. Are there some statistical analyses missing from Figure 5B and 5C?

Statistical analysis of the data is feasible only for panel A, representing the viral load. Panel B illustrates the reduction in viral load, while panel C represents the inhibition coefficient; both of these are calculated values and were derived from average viral load values. Table S4 duplicates the data from Figure 5.

  1. Supplementary files – Table S1 – What are the units for time after infection (3 or 6 what)? Table S3 – please spell out the (m±SEM) to make it clear.

The revisions have been made in accordance with the provided comments

  1. There are a few typographical errors and the odd random full stop in a strange place (such as on page 5 of the Supplementary data where the sentence starts ‘.Among Europeans, the most common………’. Please be consistent throughout.

The revisions have been made in accordance with the provided comments .

  1. Can the OD450nm please be added to the relevant Figures rather than just ‘OD450’.

The symbols in Figures 3 and 4 have been modified as requested.

Reviewer 2 Report

Comments and Suggestions for Authors

Very good work by Suvorov et al, entitled "SARS-CoV-2 Spike Protein-Expressing Enterococcus for Oral Vaccination: Immunogenicity and Protection". I am happy to accept the manuscript in its current form. Only need minor corrections like typo and grammer errors like please see line number 125, 150, 155, 202, 208 and so on.

Figures are blurred and clear resolution figures should be provide for especially Figure 3-5. Moreover, the scientific name in the captions and x-axis of Figure 3-5 should be italicized. Also the authors have put full stop after each heading. this is not in format of MDPI journals. it should be removed in revision.

Authors have mentioned clearly their objective and explained the methodology in enough details. The results and discussion sections are elaborated very well. However, conclusion can be revised as it is not well written.

Author Response

We thank the reviewer for his kind assessment of our modest work and for very useful remarks. We also enclose responses to comments.

Only need minor corrections like typo and grammer errors like please see line number 125, 150, 155, 202, 208 and so on.

We thank the reviewer for this comment. We corrected errors where we found them.

Figures are blurred and clear resolution figures should be provide for especially Figure 3-5. Moreover, the scientific name in the captions and x-axis of Figure 3-5 should be italicized. Also the authors have put full stop after each heading. this is not in format of MDPI journals. it should be removed in revision.

We thank the reviewer for this comment. We have replaced the drawings and corrected the axis labels. We also corrected the style of headings in accordance with the requirements of the journal.

Authors have mentioned clearly their objective and explained the methodology in enough details. The results and discussion sections are elaborated very well. However, conclusion can be revised as it is not well written.

We especially thank the reviewer for this remark. We rewrote the conclusion.

Reviewer 3 Report

Comments and Suggestions for Authors

This study deals with COVID-19 mucosal vaccine, an important issue for the prophylaxis of the infectious disease. The results are clearly presented, and the manuscript is well written. The authors should follow the Instructions for Authors of this journal: the abstract should be a total of about 200 words maximum and a single paragraph (252 words and 4 paragraphs in the current version).

Author Response

We thank the reviewer very much for such a positive review and useful comment. We revised the abstract according to the requirements of the journal.

Round 2

Reviewer 1 Report

Comments and Suggestions for Authors

The authors have given comprehensive replies to most of the issues raised.

I am not sure why the whole of the abstract has been highlighted as there are only 2 sentences that have been changed in the abstract, not the whole thing as is implied by the highlighting shown (unless I have missed something subtle).

There are a few issues that have not been dealt with properly.

Point 7. The authors say We have refined the manuscript's wording and made appropriate revisions.On inspection of the revised version of the manuscript that I have downloaded (V2), the section in question (now on lines 291 and 292), is identical to the original AND it still has the word etc in it. This needs to be addressed.

Line 295 has “Supplement Table 3” but line 693 has “supplementary data files.” Can these please be made consistent (ie Supplementary Table 3).

Point 15. Table 4 (between lines 351 and 352). Neither of the issues that I raised have been dealt with however the authors have stated “Table 4 has been rectified as per the recommendations provided." This is not the case on the version I have.

Point 23. The authors state “The symbols in Figures 3 and 4 have been modified as requested.

On the version I have the authors have added “nm” to Figure 4 BUT NOT to Figure 3.

Line 670 should read “In summary, the data have …….” (as opposed to the data has).

The Supplementary data Table 3 needs to be labelled as Table S3.

Comments on the Quality of English Language

The English language is fine.

Author Response

The authors especially thank the Reviewer for all the remarks and comments.

The authors have given comprehensive replies to most of the issues raised.

I am not sure why the whole of the abstract has been highlighted as there are only 2 sentences that have been changed in the abstract, not the whole thing as is implied by the highlighting shown (unless I have missed something subtle).

Response. We thank the reviewer for this question. Аccording to recommendation of other reviewer we decreased this text from 252 words to 200 and from four paragraphs to one, to follow the Instructions for Authors of this journal. We removed abstract highlighting.

Point 7 The authors say “We have refined the manuscript's wording and made appropriate revisions.” On inspection of the revised version of the manuscript that I have downloaded (V2), the section in question (now on lines 291 and 292), is identical to the original AND it still has the word etc in it. This needs to be addressed.

Response. We thank the reviewer for this remark. Yes, actually, we forgot to include the new reduction of that paragraph. The correction was done.

Line 295 has “Supplement Table 3” but line 693 has “supplementary data files.” Can these please be made consistent (ie Supplementary Table 3).

Response. We thank the reviewer for this comment. It was corrected.

Point 15 Table 4 (between lines 351 and 352). Neither of the issues that I raised have been dealt with however the authors have stated “Table 4 has been rectified as per the recommendations provided." This is not the case on the version I have.

Response. We thank the reviewer for this comment. It was corrected.

 Point 23 The authors state “The symbols in Figures 3 and 4 have been modified as requested.”

On the version I have the authors have added “nm” to Figure 4 BUT NOT to Figure 3

Response. We thank the reviewer for this remark. It was corrected.

 Line 670 should read “In summary, the data have …….” (as opposed to the data has).

Response. We thank the reviewer for this comment.  It was corrected.

The Supplementary data Table 3 needs to be labelled as Table S3.

Response. We thank the reviewer for this comment. It was corrected.